# Association Between Melatonin Use and Cataract Risk: A Target Trial Emulation Retrospective Cohort Study

**DOI:** 10.3390/antiox14081016

**Published:** 2025-08-20

**Authors:** Cheng-Hsien Hung, Jing-Yang Huang, Yu-Chien Hung, Min-Yen Hsu, James Cheng-Chung Wei

**Affiliations:** 1Department of Pharmacy, Chang Bing Show Chwan Memorial Hospital, Changhua 50544, Taiwan; chenghsien823@gmail.com; 2Institute of Medicine, Chung Shan Medical University, Taichung 40201, Taiwan; b92401086@gmail.com; 3School of Health Policy and Management, Chung Shan Medical University, Taichung 40201, Taiwan; wchinyang@gmail.com; 4School of Medicine, Chung Shan Medical University, Taichung 40201, Taiwan; 5Department of Ophthalmology, Chung Shan Medical University Hospital, Taichung 40201, Taiwan; 6Division of Allergy, Immunology and Rheumatology, Chung Shan Medical University Hospital, No. 110, Sec. 1, Jianguo N. Rd., South District, Taichung 40201, Taiwan

**Keywords:** melatonin, cataract, age-related cataract, TriNetX, cohort study, oxidative stress

## Abstract

Oxidative stress has been shown to play an important role in the development of cataracts. Melatonin is an endogenous hormone that has been proposed to have a protective effect against oxidative stress; however, its association with the risk of cataracts is uncertain. This target trial emulation study aimed to investigate the relationship between the use of melatonin and the risk of cataracts using TriNetX electronic health records from 2015 to 2023. Adults aged 40 years or older who were diagnosed with a sleep disorder and initiated treatment with either melatonin or a hypnotic benzodiazepine (BZD) were included. Patients receiving hypnotic BZD therapy constituted the active comparator group. Propensity score matching (PSM) was performed to control for covariance. Exposure was defined as the initiation of melatonin or hypnotic BZD therapy. The primary outcomes were the incidence of age-related cataracts and other cataract types, assessed using Cox proportional hazards models to estimate hazard ratios (HRs) with 95% confidence intervals (CIs). Subgroup and sensitivity analyses were conducted to reduce potential bias. After PSM, 5507 participants were included in each group. Compared with hypnotic BZD users, melatonin users were associated with a significantly lower risk of age-related cataracts (HR, 0.741; 95% CI, 0.681–0.807) and other cataracts (HR, 0.503; 95% CI, 0.433–0.584). These associations remained consistent across all subgroups and sensitivity analyses. In this target trial emulation cohort study, the use of melatonin was associated with a reduced risk of cataracts, suggesting a potential protective effect that warrants further investigation.

## 1. Introduction

Cataracts are a common ocular condition characterized by the clouding of the lens [1]. They are one of the leading causes of vision loss worldwide with a particularly significant impact on older individuals [2,3], and they can have a profound effect on quality of life [4]. Epidemiological studies have identified several major risk factors for cataracts, including age, genetic predisposition, ultraviolet (UV) exposure, and certain lifestyle factors [5,6,7]. Aging and oxidative stress have been shown to play crucial roles in the development of various diseases, including cataracts [2,8,9,10]. Oxidative stress drives cataractogenesis by promoting protein aggregation, lens opacification, and epithelial cell apoptosis. The potent antioxidant and anti-inflammatory effects of melatonin counteract these processes by scavenging reactive oxygen species and stabilizing mitochondrial function. Surgery remains the primary treatment for cataracts [11], and despite significant advances in management, they continue to represent a major public health challenge.

Melatonin is a hormone primarily produced by the pineal gland, and it has gained increasing attention for its antioxidant and anti-inflammatory properties. Its use has increased significantly among adults in the United States in recent years [12]. While melatonin is widely known for its role in regulating the sleep–wake cycle, emerging evidence suggests that it may have a protective effect against ocular diseases, including cataracts [13,14,15]. Animal studies have shown that melatonin can mitigate oxidative damage in the lens and delay cataract formation caused by factors such as UV exposure and diabetes [16,17]. Modulating the antioxidant system is also considered a promising therapeutic strategy for treating retinal degeneration [18]. Other studies have demonstrated that melatonin not only regulates sleep and ocular circadian rhythms but also plays a physiological role in the retina, ciliary body, lens, and cornea, thereby influencing ocular physiology and visual function [19,20,21]. A retrospective cohort study found that the use of melatonin supplements was associated with a reduced risk of age-related macular degeneration (AMD) in older adults, highlighting its potential application in ocular diseases [22]. In addition, a study using an AI-driven drug repurposing system suggested that melatonin may have the potential to reduce the risk of cataract surgery [23].

Despite promising preclinical evidence, studies investigating the association between the use of melatonin and risk of cataracts in humans remain limited and primarily consist of small-scale investigations or indirect evidence from related ocular conditions. To address this gap, we conducted this cohort study using a large-scale electronic health record database (TriNetX) to examine the relationship between melatonin use and cataract risk. Our findings may contribute valuable insights into the potential repurposing of melatonin as a preventive strategy for oxidative stress-related ocular conditions.

## 2. Methods

### 2.1. Data Source

This retrospective cohort study used data from the TriNetX platform, a global healthcare research network that aggregates electronic health record data from 68 U.S healthcare organizations, and includes de-identified data on diagnoses, procedures, medications, and laboratory results. It is important to note that TriNetX is not an open-access platform, and access is restricted to researchers affiliated with academic medical centers that have established agreements with TriNetX. The Western Institutional Review Board granted a waiver of informed consent for TriNetX, as the platform exclusively uses deidentified data to generate aggregated summaries and counts. This study retrospectively used de-identified data, and it was approved by the Institutional Review Board of the Affiliated Hospital of Chung Shan Medical University (CSMUH No: CS2-21176 Date 1 December 2021). The use of TriNetX for studies investigating melatonin or cataracts has previously been validated [22,24]. This study adheres to the Strengthening the Reporting of Observational Studies in Epidemiology (STROBE) guidelines [25].

### 2.2. Study Design

We used a target trial emulation design, which outlines the key elements of a hypothetical randomized controlled trial and emulates it using observational data to reduce bias and strengthen causal inference [26]. The key components of a target trial protocol compared to an emulation protocol are summarized in Appendix A. To minimize detection bias, we only included patients who had at least two healthcare visits between 1 January 2015 and 31 December 2023 for ophthalmic services or related procedures. We selected patients who used melatonin or hypnotic benzodiazepines (BZDs) after being diagnosed with a sleep disorder [27]. An active comparator group with the same indication was chosen to reduce potential selection and immortal time bias [28,29]. There is no evidence indicating an association between hypnotic BZD use and cataract development. The cohort was subsequently divided into two groups: the first group included patients aged 40 years or older who used melatonin, while the second group included patients aged 40 years or older who used hypnotic BZD. Figure 1 shows the study enrollment flowchart. The index date was defined as the date of the first prescription of melatonin or a hypnotic BZD after the diagnosis of a sleep disorder. To further ensure adequate exposure, we excluded individuals who did not receive a second prescription for melatonin or hypnotic BZD at least three months after the initial prescription [30]. This criterion, informed by prior pharmacoepidemiologic studies, was designed to minimize the inclusion of sporadic users and better capture those with sustained or repeated use [31]. While it does not confirm daily adherence, it offers a reasonable proxy for long-term exposure during the observation period and helps exclude patients with only short-term use. Patients with a pre-existing diagnosis of lens disease prior to the index date were also excluded. All queries used in the data analysis were based on the International Classification of Diseases and related health issues, with the relevant codes provided in Appendix A. The schematic overview of the emulated trial is presented in Appendix A.

### 2.3. Statistical Analysis

Baseline patient information and comorbidities were obtained from the year prior to the index date. To balance differences in baseline characteristics between the two groups, propensity score matching (PSM) was applied to match age, race, gender, body mass index (BMI), comorbidities, and medications in a 1:1 ratio. The matching process was performed using the built-in functionality of the TriNetX platform. All data were reported by healthcare organizations in collaboration with the TriNetX platform. We used the standardized mean difference (SMD) to assess the balance of baseline characteristics between the matched cohorts. Variables with an SMD < 0.1 were considered well matched. A detailed list of covariates included in the PSM is presented in Appendix A. To avoid protopathic and detection bias, we implemented a one-year lag time for analysis [32]. Kaplan–Meier survival analysis was used to assess the incidence rates of outcomes over one to five years, with statistical significance defined as two-sided *p*-values < 0.05. Cox proportional hazards models were used to calculate adjusted hazard ratios (aHRs) with 95% confidence intervals (CIs). The primary outcomes were the incidence of age-related cataracts and other types of cataracts, and the secondary outcomes encompassed various cataract subtypes. Considering the potential moderation effects of gender, ethnicity and age, subgroup analyses were conducted based on these factors. We also performed lag-time sensitivity analysis to examine the association between melatonin use and cataract risk. By applying different lag times (e.g., no lag, 1 year, or 2 years) between the index date and cataract outcome, the analysis aimed to minimize the potential influence of protopathic bias. We conducted a sensitivity analysis focusing on the non-melatonin control group. In this analysis, the index date for the melatonin group was defined as the date of the first melatonin prescription concurrent with an ophthalmology visit, while the index date for the control group was set as the date of their first ophthalmology visit. In addition, urinary tract infection was selected as a negative outcome for sensitivity analysis to assess the robustness of the model.

## 3. Results

### 3.1. Baseline Characteristics of the Study Subjects

Before PSM, the melatonin group consisted of 5670 patients with a mean age of 61.3 ± 12.6 years, while the hypnotic BZD group included 51,711 patients with a mean age of 57.2 ± 9.5 years. In terms of sex distribution, 60.3% of the melatonin group were females, compared to 56.9% in the hypnotic BZD group. Regarding race, 66.7% of the melatonin group were White, 16.1% Black or African American, and 2.5% Asian, compared to 69.5%, 16.1%, and 1.8% in the hypnotic BZD group. After matching, each group contained 5507 patients, and the baseline characteristics were well balanced. The characteristics of the study participants are summarized in Table 1.

### 3.2. Primary Outcomes

Table 2 shows the number of patients with cataracts in the melatonin and hypnotic BZD groups, along with the 5-year aHRs for the incidence of cataracts in the melatonin users compared with the hypnotic BZD users. The Kaplan–Meier curve of the cumulative probability of the risk of cataracts is presented in Figure 2. After 5 years of follow-up, the melatonin users had a lower risk of age-related cataracts (HR 0.741, 95% CI: 0.681–0.807). In addition, the incidence of other cataracts was lower in the melatonin group compared to the hypnotic BZD group (HR 0.503, 95% CI 0.433–0.584).

### 3.3. Subgroup Analysis

Subgroup analysis showed that the use of melatonin was associated with a significantly lower risk of cataracts across gender, age, and racial groups. For age-related cataracts, the reduction in risk was slightly greater in females (HR 0.716, 95% CI 0.639–0.802) compared to males (HR 0.804, 95% CI 0.698–0.927). However, the difference between the two groups was not significant (*p* for interaction = 0.21). Among the different racial groups, a substantial reduction was shown in the White patients (HR 0.745, 95% CI 0.670–0.830), while no significant reductions were observed in the Black (HR 0.896, 95% CI 0.733–1.095) or Asian (HR 0.713, 95% CI 0.415–1.224) patients. The difference between the groups was not significant (*p* for interaction = 0.27). A significant risk reduction was noted in the older adults (≥65 years) (HR 0.856, 95% CI 0.749–0.979), but no significant effect was observed in the younger adults (40–64 years, HR 0.919, 95% CI 0.819–1.032). The difference between these age groups was not significant (*p* for interaction = 0.43) (Figure 3A). For other types of cataracts, both males and females had similar risk reductions, with males having an HR of 0.46 (95% CI 0.35–0.59) and females an HR of 0.47 (95% CI 0.38–0.57). This difference was not significant (*p* for interaction = 0.89). White and Black patients showed significant risk reductions (HR 0.548, 95% CI 0.452–0.664; HR 0.612, 95% CI 0.416–0.899, respectively), while the results in the Asian patients were inconclusive (HR 0.843, 95% CI 0.325–2.183). The difference between the groups was not significant (*p* for interaction = 0.63). Both younger adults (40–64 years, HR 0.473, 95% CI 0.374–0.600) and older adults (≥65 years, HR 0.629, 95% CI 0.519–0.762) showed significant reductions. The difference between the groups was not significant (*p* for interaction = 0.07) (Figure 3B).

### 3.4. Secondary Outcomes

In the analysis of different types of cataracts, melatonin use was statistically significantly associated with a reduced risk of secondary cataracts (HR 0.428, 95% CI 0.321–0.572) and unspecified cataracts (HR 0.498, 95% CI 0.417–0.594). However, for rarer types of cataracts, such as complicated cataracts, traumatic cataracts, and drug-induced cataracts, the results were inconclusive due to the limited number of cases. Detailed risks of the study events are presented in Table 2.

### 3.5. Additional Analysis

In the lag-time sensitivity analysis, melatonin use was consistently associated with a reduced incidence of cataracts across different lag times. For age-related cataracts, melatonin use was associated with a significant reduction in risk: with no lag time (HR 0.796, 95% CI 0.741–0.855), 1-year lag (HR 0.741, 95% CI 0.681–0.807), and 2-year lag (HR 0.826, 95% CI 0.742–0.919). For other cataracts, melatonin use demonstrated a consistent effect: with no lag time (HR 0.555, 95% CI 0.489–0.629), 1-year lag (HR 0.503, 95% CI 0.433–0.584), and 2-year lag (HR 0.542, 95% CI 0.448–0.656). Detailed results are presented in Appendix A. In the sensitivity analysis comparing melatonin users to non-users, melatonin use was associated with a significantly lower risk of both age-related cataracts (HR 0.785, 95% CI 0.735–0.839) and other cataracts (HR 0.885, 95% CI 0.826–0.948). Detailed results are presented in Appendix A. Analysis of the negative outcome showed no significant difference in the risk of urinary tract infection (HR 0.962, 95% CI 0.825–1.121) between the two groups, indicating that melatonin use did not increase the risk of these events. Detailed results are presented in Table 2.

## 4. Discussion

The results of this target trial emulation study showed that the patients who used melatonin had a lower risk of cataract development compared to those who used a hypnotic BZD. The incidence of age-related cataracts was significantly lower in the melatonin group compared to the hypnotic BZD group (HR 0.741, 95% CI 0.681–0.807), and the incidence of other types of cataracts was also significantly lower in the melatonin group (HR 0.503, 95% CI 0.433–0.584). This association remained consistent across different cataract subtypes and in subgroup analysis of age, sex, and race. These findings provide preliminary evidence supporting the potential protective role of melatonin in reducing the risk of cataract development.

Our findings are consistent with previous animal studies which showed that melatonin could slow the oxidation of lens proteins. One study found that melatonin protected the lenses of rats exposed to UVB radiation from oxidative damage and delayed cataract formation, and that melatonin inhibited ferroptosis by regulating the SIRT6/p-Nrf2/GPX4 and SIRT6/NCOA4/FTH1 signaling pathways [16]. These findings suggest that melatonin may have potential in treating or improving diseases related to ferroptosis, such as cataracts. In another study, Khorsand et al. reported that melatonin could slow cataract formation and progression in diabetic rats [33]. Melatonin has also been shown to have an antioxidant effect against radiation-induced cataracts by increasing the activity of superoxide dismutase (SOD) and glutathione peroxidase (GSH-Px) while decreasing the levels of lipid peroxidation (as assessed using malondialdehyde [MDA]), thereby reducing oxidative stress caused by radiation and protecting the lens from radiation-induced damage [34]. Yağci et al. further showed that melatonin had a protective effect against sodium selenite-induced cataract formation in rats by significantly reducing oxidative stress markers (MDA, protein carbonyl, xanthine oxidase) and enhancing antioxidant enzyme levels (SOD, catalase), emphasizing its potential as an endogenous antioxidant and ability to prevent cataracts [35]. In addition, a recent real-world study found that melatonin was associated with a reduced risk of the development and progression of AMD (RR = 0.42; 95% CI = 0.28–0.62), suggesting the potential of melatonin as a preventive therapy for age-related ocular diseases [22]. Although we did not include biochemical markers in the present study due to limitations of the database, preclinical studies have also demonstrated that melatonin reduces oxidative stress in the lens by decreasing MDA levels and enhancing antioxidant enzymes such as SOD and GSH-Px [16,17,35]. In addition, melatonin has been shown to regulate Nrf2-mediated antioxidant responses and inhibit NLRP3 inflammasome activation in lens epithelial cells [36]. These findings underscore the potential protective role of melatonin against cataractogenesis. Furthermore, clinical evidence linking melatonin to a reduced risk of AMD also suggests a possible benefit in oxidative stress-related ocular diseases.

Further investigations are still required to elucidate the mechanisms underlying the potential role of melatonin in preventing cataracts. Melatonin plays a crucial role in ocular health by regulating angiogenesis and maintaining blood-retinal barrier integrity. Melatonin also supports retinal cell health by modulating autophagy through the Sirt1/mTOR pathway, reducing inflammation, promoting antioxidant enzyme activity, and stabilizing intraocular pressure. In addition, melatonin has been shown to protect retinal ganglion cells by influencing aging and inflammatory pathways [37]. Previous research has highlighted the protective effects of melatonin against oxidative stress and inflammation induced by hydrogen peroxide and white light-emitting diode light in lens epithelial cells. Melatonin reduces reactive oxygen species (ROS) generation, enhances antioxidant activity, and inhibits NLRP3 (nucleotide-binding oligomerization domain-like receptor family, pyrin domain-containing 3) inflammasome activation, demonstrating its potential as a therapeutic agent for cataract prevention and management [36].

This study has several strengths. First, it used electronic health record data from the TriNetX platform, which includes data from multiple healthcare institutions, thereby enhancing the robustness of the analysis. Second, PSM was performed to effectively control for baseline characteristics including age, sex, BMI, and comorbidities, minimizing potential confounding bias. Third, patients receiving hypnotic BZDs were chosen as the active comparator group, all of whom were diagnosed with a sleep disorder prior to prescription, thereby reducing indication and time-related biases. Fourth, we conducted subgroup analyses across different populations and obtained consistent results, further enhancing the conclusions. In addition, the use of urinary tract infection as a negative control outcome adds to the robustness of the findings. Finally, we emulated a target trial to minimize bias and rigorously assess the association between the use of melatonin and risk of cataracts.

However, some limitations must be acknowledged. First, as an observational study, it may still be subject to coding errors or confounding bias. Although we selected an active comparator group and performed PSM to minimize these influences, the possibility of residual bias cannot be entirely ruled out, especially from unmeasured factors such as UV exposure. Second, due to limitations of the database, we were unable to assess the dosage, formulation and duration of melatonin use, and we were also unable to evaluate medication adherence. These limitations may have resulted in exposure misclassification and potentially biased the estimated effects. Future studies should capture detailed treatment information to confirm our findings. Third, although we observed associations between the use of melatonin and a lower incidence of different cataract subtypes, the limited number of events in the analysis may have affected the strength of the results. Fourth, because the study relied on electronic health record data from a specific network, the findings may not be applicable to other populations. Fourth, due to the limitations of the TriNetX database, we were unable to adjust for visual acuity, history of ophthalmologic surgeries, or the frequency of ophthalmologic examinations. This limitation may have influenced diagnostic accuracy and effect estimates. Fifth, cataract outcomes were based on ICD-10 diagnosis codes, which help capture non-surgical cases but may not fully reflect disease severity or progression. Lastly, as an observational study, our findings reflect an association rather than a causal relationship.

Future research may consider evaluating the long-term effects of melatonin on cataract risk through prospective cohort studies or randomized controlled trials. Investigations into the optimal dosage, duration of use, and adherence patterns would help clarify its potential therapeutic role and improve the generalizability of findings for clinical application.

## 5. Conclusions

The results of this study demonstrated an association between melatonin use and reduced cataract risk. Although the possibility of residual confounding remains, these findings provide a basis for future research to clarify the potential role of melatonin in cataract prevention.

## Figures and Tables

**Figure 1 antioxidants-14-01016-f001:**
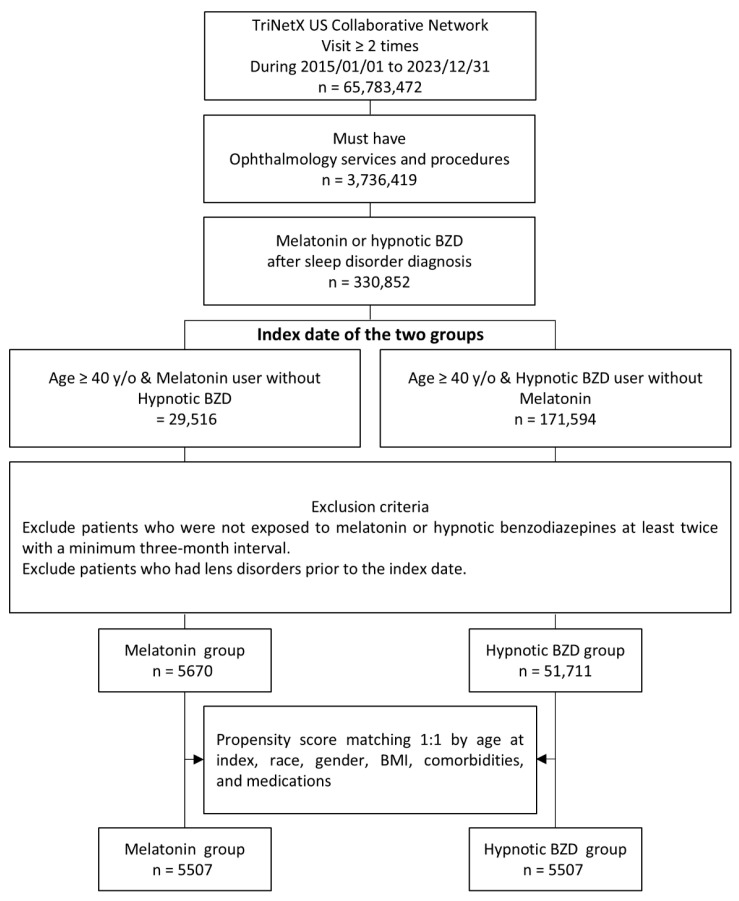
Study enrollment flowchart.

**Figure 2 antioxidants-14-01016-f002:**
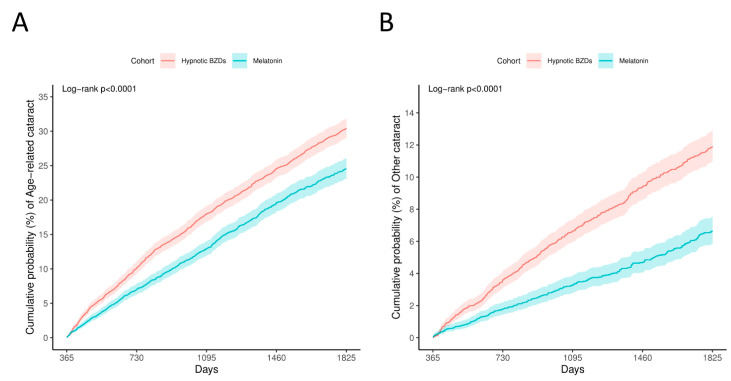
Kaplan–Meier curves of the cumulative probability of (**A**) Age-Related Cataracts (**B**) Other Cataracts.

**Figure 3 antioxidants-14-01016-f003:**
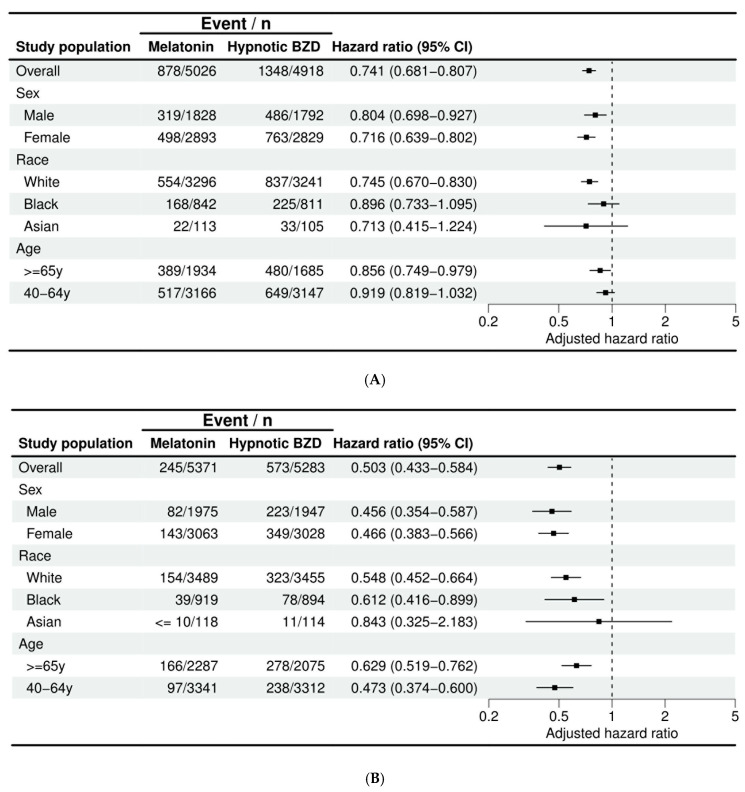
Subgroup analysis for risk of (**A**) Age-Related Cataract (**B**) Other Cataract.

**Table 1 antioxidants-14-01016-t001:** Baseline characteristics of the participants before and after propensity score matching (PSM).

	Before PSM			After PSM		
	Melatonin	Hypnotic BZD	SMD	Melatonin	Hypnotic BZD	SMD
N	5670	51711		5507	5507	
Age at index (year)	61.3 ± 12.6	57.2 ± 9.5	0.3675	60.9 ± 12.5	60.3 ± 10.1	0.0527
Female sex, n (%)	3417 (60.3%)	29,412 (56.9%)	0.0688	3327 (60.4%)	3462 (62.9%)	0.0504
Race, n (%)						
White	3784 (66.7%)	35,959 (69.5%)	0.0601	3673 (66.7%)	3821 (69.4%)	0.0577
Black or African American	911 (16.1%)	8336 (16.1%)	0.0015	890 (16.2%)	813 (14.8%)	0.0387
Asian	140 (2.5%)	907 (1.8%)	0.0498	135 (2.5%)	120 (2.2%)	0.0181
Body mass index (BMI), n (%)						
<30	2459 (43.4%)	17,608 (34.1%)	0.1922	2350 (42.7%)	2410 (43.8%)	0.0220
30–40	1855 (32.7%)	20,132 (38.9%)	0.1299	1827 (33.2%)	1823 (33.1%)	0.0015
≧40	747 (13.2%)	9627 (18.6%)	0.1493	745 (13.5%)	704 (12.8%)	0.0220
Lifestyle or environmental factor, n (%)						
Nicotine dependence	757 (13.4%)	6440 (12.5%)	0.0268	746 (13.5%)	726 (13.2%)	0.0107
Alcohol related disorders	390 (6.9%)	1507 (2.9%)	0.1845	376 (6.8%)	396 (7.2%)	0.0142
Comorbidity, n (%)						
Diseases of the circulatory system	3952 (69.7%)	36,861 (71.3%)	0.0347	3829 (69.5%)	3695 (67.1%)	0.0523
Hypertensive diseases	3354 (59.2%)	31,487 (60.9%)	0.0355	3252 (59.1%)	3094 (56.2%)	0.0581
Diabetes mellitus	1723 (30.4%)	15,591 (30.2%)	0.0052	1678 (30.5%)	1600 (29.1%)	0.0310
Type 2 diabetes mellitus with ophthalmic complications	145 (2.6%)	1210 (2.3%)	0.0141	142 (2.6%)	118 (2.1%)	0.0287
Cerebrovascular diseases	609 (10.7%)	3294 (6.4%)	0.1567	574 (10.4%)	591 (10.7%)	0.0100
Diseases of arteries, arterioles and capillaries	518 (9.1%)	4591 (8.9%)	0.0090	500 (9.1%)	491 (8.9%)	0.0057
Neoplasms	1238 (21.8%)	18,424 (35.6%)	0.3084	1216 (22.1%)	1158 (21.0%)	0.0256
Insomnia	2126 (37.5%)	11,231 (21.7%)	0.3509	2036 (37.0%)	2086 (37.9%)	0.0188
Anxiety, dissociative, stress-related, somatoform and other nonpsychotic mental disorders	2016 (35.6%)	13,562 (26.2%)	0.2029	1939 (35.2%)	1993 (36.2%)	0.0205
Depressive episode	1760 (31.0%)	12,760 (24.7%)	0.1423	1679 (30.5%)	1662 (30.2%)	0.0067
Epilepsy and recurrent seizures	217 (3.8%)	1057 (2.0%)	0.1058	208 (3.8%)	185 (3.4%)	0.0225
Parkinson’s disease	247 (4.4%)	224 (0.4%)	0.2587	159 (2.9%)	164 (3.0%)	0.0054
Alzheimer’s disease	108 (1.9%)	48 (0.1%)	0.1830	46 (0.8%)	40 (0.7%)	0.0124
Medications, n (%)						
Benzodiazepine related drugs	691 (12.2%)	6722 (13.0%)	0.0245	685 (12.4%)	683 (12.4%)	0.0011
Other hypnotics and sedatives	113 (2.0%)	3960 (7.7%)	0.2667	112 (2.0%)	106 (1.9%)	0.0078

**Table 2 antioxidants-14-01016-t002:** The risk of study events within 5 years after the index date among the study cohort.

	Event Number		
Outcome	Melatonin Cohort (*n* = 5507)	Hypnotic BZD Cohort (*n* = 5507)	Hazard Ratio (95% CI)
Age-related cataract	876	1348	0.741, (0.681, 0.807)
Other cataract	245	573	0.503, (0.433, 0.584)
Traumatic cataract	≤10	≤10	0.840, (0.140, 5.045)
Complicated cataract	≤10	≤10	0.890, (0.148, 5.339)
Drug induced cataract	0	≤10	NA
Secondary cataract	62	178	0.428, (0.321, 0.572)
Other specified cataract	27	41	0.804, (0.495, 1.309)
Unspecified cataract	175	418	0.498, (0.417, 0.594)
Negative outcome			
Urinary tract infection	284	391	0.962, (0.825, 1.121)

## Data Availability

The original contributions presented in this study are included in the article/Appendix A. Further inquiries can be directed to the corresponding authors.

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
