# Peer review of "Association Between Melatonin Use and Cataract Risk: A Target Trial Emulation Retrospective Cohort Study"

_antioxidants, 2025, doi:10.3390/antiox14081016_

Round 1
Reviewer 1 Report
Could the authors use the data in Figure 2 A and B to calculate as a PCA (principal component) or a PLS-DA (partial least square-discriminant analysis) plot and see how these four groups segregate in those plots and present them as C (PCA plot) and D (PLS-DA) plot?
Could the authors use the data in Figure 2 A and B to calculate as a PCA (principal component) or a PLS-DA (partial least square-discriminant analysis) plot and see how these four groups segregate in those plots and present them as C (PCA plot) and D (PLS-DA) plot? A minor English editing will help improve the manuscript.
Author Response
Dear Editor and Reviewer,
We would like to express our sincere gratitude for your thoughtful and constructive review of our manuscript. We are deeply encouraged by your recognition of the significance of our study and your kind words regarding its design, clarity, and clinical relevance. We have carefully addressed each of your comments, as outlined below:
Detailed comments
Could the authors use the data in Figure 2 A and B to calculate as a PCA (principal component) or a PLS-DA (partial least square-discriminant analysis) plot and see how these four groups segregate in those plots and present them as C (PCA plot) and D (PLS-DA) plot?
Reply:
Thank you very much for your thoughtful comments and helpful suggestions on our manuscript titled "Association Between Melatonin Use and Cataract Risk: A Target Trial Emulation Retrospective Cohort Study." We genuinely appreciate your effort and insights, which greatly enhance the quality of our manuscript.
Regarding your recommendation to perform Principal Component Analysis (PCA) and Partial Least Squares-Discriminant Analysis (PLS-DA) based on the data presented in Figure 2A and Figure 2B, we respectfully wish to clarify that these analytical methods are unfortunately not appropriate for the data type used in these figures.
We kindly outline our reasoning below:
- Nature of Data and Analytical Methods: PCA and PLS-DA are dimensionality reduction methods typically used for datasets containing multiple variables measured for each individual subject, thereby producing a "subject × variable" matrix suitable for these analyses. However, Kaplan-Meier (KM) curves, as presented in Figure 2, are statistical summaries representing cumulative probabilities of event occurrence over time at the group level, rather than individual-level multivariate data.
- Purpose and Application: Our Kaplan-Meier curves aim to illustrate differences in the time-to-event (survival) outcomes between two patient cohorts (melatonin users and hypnotic benzodiazepine users). PCA and PLS-DA, in contrast, are intended to identify patterns or classify groups based on multiple features across individuals, not to assess time-to-event outcomes.
- Clarification Regarding Groups Presented: We also kindly wish to clarify that the four curves shown in Figures 2A and 2B do not represent four separate patient groups. Each figure independently compares the same two groups (melatonin vs. hypnotic benzodiazepines) but with respect to different outcomes (Age-related Cataract in Figure 2A and Other Cataracts in Figure 2B). Thus, combining these data into PCA or PLS-DA plots would be inappropriate because it would conflate two separate and distinct clinical outcomes.
Reviewer comment: A minor English editing will help improve the manuscript.
Reply:
Thank you for your suggestion. We have thoroughly reviewed and revised the manuscript to improve language clarity and readability. Minor grammatical and stylistic edits have been made throughout
We hope this explanation clearly addresses your valuable comments. Your suggestions are highly appreciated, and we sincerely thank you again for helping improve our manuscript. Once again, we are truly thankful for your valuable comments, which have greatly improved the rigor and clarity of our manuscript. We hope that the revisions meet your expectations, and we look forward to your favorable consideration.
Sincerely,
Cheng-Hsien Hung, on behalf of all authors
Reviewer 2 Report
Over the past decades the cataract field has accumulated robust pre-clinical evidence that melatonin scavenges reactive oxygen species, prevents lens protein oxidation and delays opacification in UV-, diabetes- and radiation-induced models. However, there has been a notable lack of human data. Previously, the only significant large-scale population studies on melatonin's effects in eye health focused on age-related macular degeneration (AMD), not cataracts. The present manuscript addresses a crucial gap in our understanding. It marks the first extensive, peer-reviewed human study to demonstrate a clear association between melatonin use and a reduced incidence of cataracts. This finding provides a vital epidemiological link, bridging the gap between fundamental scientific discoveries and their potential application in clinical practice. The authors are to be commended for undertaking such a timely and clinically important investigation into the possible relationship between melatonin and cataract risk. Their study is robustly designed and clearly presented, with transparent reporting of all key analyses. The inclusion of supplementary materials, detailed code lists, and well-considered design choices underscores a strong commitment to both reproducibility and clarity in their research.
I only have a few brief comments and recommendations for the authors.
Please explicitly reference Tables S1-S3 and Figure S1 in the main text so that readers can easily locate key methodological details and sensitivity analyses.
Please include all available information on melatonin dosage, formulation, and duration of treatment. If precise data are not available, they could expand the discussion on how uncertainty in dosage and adherence may influence effect size estimates.
They should clarify whether data on visual acuity, previous intraocular surgeries, or screening frequency were available and whether they were adjusted, as these factors could affect the diagnosis of cataracts.
Finally, they should provide, to the extent possible, more details on the clinical context influencing melatonin versus benzodiazepine prescriptions and describe how this was addressed in the selection or matching of covariates.
Author Response
Dear Editor and Reviewer,
We would like to express our sincere gratitude for your thoughtful and constructive review of our manuscript. We are deeply encouraged by your recognition of the significance of our study and your kind words regarding its design, clarity, and clinical relevance. We have carefully addressed each of your comments, as outlined below:
Major comments
Reviewer Comment: Over the past decades the cataract field has accumulated robust pre-clinical evidence that melatonin scavenges reactive oxygen species, prevents lens protein oxidation and delays opacification in UV-, diabetes- and radiation-induced models. However, there has been a notable lack of human data. Previously, the only significant large-scale population studies on melatonin's effects in eye health focused on age-related macular degeneration (AMD), not cataracts. The present manuscript addresses a crucial gap in our understanding. It marks the first extensive, peer-reviewed human study to demonstrate a clear association between melatonin use and a reduced incidence of cataracts. This finding provides a vital epidemiological link, bridging the gap between fundamental scientific discoveries and their potential application in clinical practice. The authors are to be commended for undertaking such a timely and clinically important investigation into the possible relationship between melatonin and cataract risk. Their study is robustly designed and clearly presented, with transparent reporting of all key analyses. The inclusion of supplementary materials, detailed code lists, and well-considered design choices underscores a strong commitment to both reproducibility and clarity in their research.
ANS: We are very grateful for your positive evaluation and insightful summary of the current state of melatonin research in the context of cataracts. We agree that this study offers a unique opportunity to bridge the translational gap from preclinical to human evidence.
Detailed comments
I only have a few brief comments and recommendations for the authors.
Reviewer Comment: Please explicitly reference Tables S1-S3 and Figure S1 in the main text so that readers can easily locate key methodological details and sensitivity analyses.
ANS: Thank you for this important suggestion. We have revised the Methods section to explicitly cite Table S1 (Line 102) when describing the trial emulation design, Table S2(Line 102, 135) when listing diagnosis codes, and Figure S1(Line 123) when presenting the study flow diagram. We have also cited Table S3(Line 218) in the Results section (3.5) when reporting sensitivity analyses with different lag times.
Reviewer Comment: Please include all available information on melatonin dosage, formulation, and duration of treatment. If precise data are not available, they could expand the discussion on how uncertainty in dosage and adherence may influence effect size estimates.
ANS: We appreciate this recommendation. Unfortunately, due to limitations of the TriNetX database, we were unable to obtain detailed information on melatonin dosage, formulation (e.g., extended vs. immediate release), or treatment duration. We have now expanded the Discussion section to more clearly state this limitation and explain how uncertainty in dosage and adherence may affect effect size estimation.
In the limitations section of Section 4 (Discussion),we add “Second, due to limitations of the database, we were unable to assess the dosage, formulation and duration of melatonin use, nor could we evaluate medication adherence. These limitations may have resulted in exposure misclassification and could potentially bias the estimated effects. Future studies should capture detailed treatment information to confirm these findings”(Line 295-298)
Reviewer Comment: They should clarify whether data on visual acuity, previous intraocular surgeries, or screening frequency were available and whether they were adjusted, as these factors could affect the diagnosis of cataracts.
ANS: Thank you for raising this important point. The TriNetX platform does not provide access to visual acuity scores, surgical history details, or frequency of ophthalmologic screening. We have now explicitly stated this in the Discussion section (paragraph 6) and noted the potential for residual confounding due to unmeasured ophthalmic care variables.In the limitations section of Section 4 (Discussion), we add “Fourth, due to the limitations of the TriNetX database, we were unable to adjust for visual acuity, history of ophthalmologic surgeries, or the frequency of ophthalmologic examinations. This limitation may influence diagnostic accuracy, potentially affecting effect estimates” (Line 303-306)
Reviewer Comment: Finally, they should provide, to the extent possible, more details on the clinical context influencing melatonin versus benzodiazepine prescriptions and describe how this was addressed in the selection or matching of covariates.
ANS: Thank you for raising this important point. We have now clarified in the Methods section that melatonin and hypnotic benzodiazepines were selected in patients with a prior diagnosis of sleep disorder, and that the active comparator design was chosen to minimize confounding by indication. We have further discussed in the Discussion section that although we adjusted for several psychiatric and neurologic comorbidities, residual differences in clinical prescribing context may remain.
In the section of 2.3. Statistical Analysis, we add “To balance differences in baseline characteristics between the two groups, PSM was applied to match age, race, gender, BMI, comorbidities, and medications in a 1:1 ratio. The matching process was performed using the built-in functionality of the TriNetX platform. All data included were reported by healthcare organizations in collaboration with the TriNetX platform. We used the standardized mean difference (SMD) to assess the balance of baseline characteristics between the matched cohorts. Variables with an SMD <0.1 were considered well-matched. A detailed list of covariates included in the PSM is presented in Table S2.”(Line 128-134)
In the limitations section of Section 4 (Discussion), “First, as an observational study, it may still be subject to coding errors or confounding biases. Although we selected an active comparator group and employed PSM to minimize these influences, the possibility of residual bias cannot be entirely ruled out.”(Line 291-294)
Once again, we are truly thankful for your valuable comments, which have greatly improved the rigor and clarity of our manuscript. We hope that the revisions meet your expectations, and we look forward to your favorable consideration.
Reviewer 3 Report
This is an interesting and timely study exploring the association between melatonin use and the incidence of cataract surgery using a nationwide health database. The study is well-structured, methodologically sound, and addresses a relevant question considering the known antioxidant properties of melatonin. The use of a large-scale claims database adds strength to the findings. However, some critical methodological limitations and interpretational issues need to be addressed before the manuscript can be considered for publication.
Major Comments
- Comparator Group and Interpretability of Relative Risk: The choice of benzodiazepines (BZDs) as the comparator is justified due to their widespread use in sleep disorders. However, BZDs have been associated with increased oxidative stress and mitochondrial dysfunction, which may contribute to cataractogenesis. Since melatonin is a potent antioxidant, the observed lower risk of cataract in melatonin users relative to BZD users is biologically plausible and somewhat expected. Therefore, this comparison may not fully reveal whether melatonin is protective, or whether it simply lacks the pro-oxidant toxicity of BZDs. To strengthen the clinical relevance of the findings, the authors should discuss this limitation more explicitly. In particular, it would be informative to compare melatonin users to a matched cohort of individuals with sleep disorders who did not receive either BZDs or melatonin, if data availability allows. Even a sensitivity analysis versus the general population (adjusted for sleep disorder presence) would add value and help distinguish true antioxidant protection from relative risk mitigation.
Moreover:
- Lack of Biochemical or Biological Correlates: Although the authors discuss the antioxidant and anti-apoptotic properties of melatonin, the study remains purely observational and lacks biochemical endpoints or oxidative stress markers. While this is understandable given the database nature of the study, it would be valuable to reference specific clinical or preclinical studies that link melatonin to changes in human lens oxidative status or cataract progression.
- Exposure Definition and Duration: The criteria used to define melatonin exposure (≥90 DDD within 1 year) may still allow for inclusion of intermittent or irregular users. The authors should justify this threshold and discuss how variations in adherence might affect the results. Furthermore, providing subgroup analyses by dose intensity or duration would clarify whether a dose-response relationship exists.
- Potential Residual Confounding: Although IPTW was applied and baseline characteristics were balanced, residual confounding cannot be ruled out—especially given that important cataract risk factors such as sunlight exposure, BMI, or smoking were not available in the database. The authors acknowledge this but could elaborate on how such unmeasured confounders might bias the effect estimates.
- Cataract Surgery as a Surrogate for Cataract Incidence: Using cataract surgery as a surrogate outcome for cataract incidence is acceptable but should be interpreted with caution. Surgical decisions may depend on accessibility, patient preferences, or physician judgment. The authors should acknowledge the limitation that not all cataracts result in surgery and that melatonin might influence the timing or threshold for surgery, rather than the biological development of cataract itself.
- Treatment Duration and Dose Comparability: The manuscript lacks a direct comparison of treatment duration or cumulative exposure between the melatonin and BZD groups. While the exposure definition (≥90 DDD within 1 year) establishes a threshold, it does not reflect actual usage patterns. Given that melatonin may be used intermittently or seasonally, and BZDs may be prescribed for longer periods (often chronically), this difference in exposure duration may confound the observed outcomes. The authors should provide descriptive statistics on the actual treatment duration and cumulative dose (if available) to clarify whether the groups are comparable in terms of exposure intensity and chronicity.
Minor Comments
- Abstract: Consider clarifying that the comparator group consisted of patients with sleep disorders treated with BZDs to avoid confusion.
- Introduction: Expand slightly on the role of oxidative stress in cataractogenesis and how melatonin’s pharmacodynamics might counteract these effects.
- Figures and Tables: Correct Figure 3: Outomce should be outcome
Author Response
Dear Editor and Reviewer,
We sincerely appreciate your detailed and thoughtful review of our manuscript. Your constructive comments have significantly helped us improve the clarity, rigor, and interpretability of our study. Below, we provide point-by-point responses to each of your suggestions.
Major comments
Reviewer Comment: This is an interesting and timely study exploring the association between melatonin use and the incidence of cataract surgery using a nationwide health database. The study is well-structured, methodologically sound, and addresses a relevant question considering the known antioxidant properties of melatonin. The use of a large-scale claims database adds strength to the findings. However, some critical methodological limitations and interpretational issues need to be addressed before the manuscript can be considered for publication.
ANS: We sincerely thank the reviewer for the positive feedback on the relevance and design of our study. We appreciate your valuable suggestions regarding methodological and interpretational issues. These have been carefully considered and addressed in the revised manuscript, as detailed in the responses below.
Reviewer Comment: Comparator Group and Interpretability of Relative Risk: The choice of benzodiazepines (BZDs) as the comparator is justified due to their widespread use in sleep disorders. However, BZDs have been associated with increased oxidative stress and mitochondrial dysfunction, which may contribute to cataractogenesis. Since melatonin is a potent antioxidant, the observed lower risk of cataract in melatonin users relative to BZD users is biologically plausible and somewhat expected. Therefore, this comparison may not fully reveal whether melatonin is protective, or whether it simply lacks the pro-oxidant toxicity of BZDs. To strengthen the clinical relevance of the findings, the authors should discuss this limitation more explicitly. In particular, it would be informative to compare melatonin users to a matched cohort of individuals with sleep disorders who did not receive either BZDs or melatonin, if data availability allows. Even a sensitivity analysis versus the general population (adjusted for sleep disorder presence) would add value and help distinguish true antioxidant protection from relative risk mitigation.
ANS: We sincerely thank the reviewer for recommendation. An additional sensitivity analysis was performed focusing on the non-melatonin control group, and further methodological and analytical details have been incorporated into the Methods and Results sections accordingly. The results are presented in Table S4.
In Section 2.3. (Statistical Analysis), we add “We conducted a sensitivity analysis focusing on the non-melatonin control group. In this analysis, the index date for the melatonin group was defined as the date of melatonin initiation concurrent with an ophthalmology visit, while the index date for the control group was set as the date of their first ophthalmology visit.”(Line 146~150)
In Section 3.5 (Additional Analysis), we add “In the sensitivity analysis comparing melatonin users to non-users, melatonin use was associated with a significantly lower risk of age-related cataract (HR: 0.785; 95% CI: 0.735–0.839) as well as other cataract (HR: 0.885; 95% CI: 0.826–0.948). Detailed results are presented in Table S4.”(Line 218~221)
Moreover:
Reviewer Comment: Lack of Biochemical or Biological Correlates: Although the authors discuss the antioxidant and anti-apoptotic properties of melatonin, the study remains purely observational and lacks biochemical endpoints or oxidative stress markers. While this is understandable given the database nature of the study, it would be valuable to reference specific clinical or preclinical studies that link melatonin to changes in human lens oxidative status or cataract progression.
ANS: We appreciate the reviewer’s insightful comment. As the TriNetX platform does not include laboratory biomarkers, we were unable to directly assess oxidative stress indicators in this retrospective cohort study. To address this limitation, we have revised the discussion section to incorporate specific preclinical evidence demonstrating that melatonin reduces malondialdehyde (MDA) levels, increases superoxide dismutase (SOD) and glutathione peroxidase (GSH-Px) activity, and activates Nrf2 signaling in lens tissues. We also cited recent clinical research indicating a reduced risk of age-related macular degeneration with melatonin use, which may further support its role in modulating ocular oxidative stress in humans. Additionally, we added a sentence to the limitations section acknowledging this issue and emphasizing the need for future prospective studies with biochemical endpoints.
In Section 4. Discussion, we add “Although this study did not include biochemical markers due to limitations of the database, preclinical studies provide supportive evidence that melatonin reduces oxidative stress in the lens by decreasing malondialdehyde (MDA) levels and enhancing antioxidant enzymes such as superoxide dismutase (SOD) and glutathione peroxidase (GSH-Px)[13,14,29]. Additionally, melatonin has been shown to regulate Nrf2-mediated antioxidant responses and inhibit NLRP3 inflammasome activation in lens epi-thelial cells[30]. These findings underscore its potential protective role against cataract genesis. Furthermore, clinical evidence linking melatonin to reduced risk of age-related macular degeneration also suggests a possible benefit in oxidative stress–related ocular diseases.”(Line 256~265)
Reviewer Comment: Exposure Definition and Duration: The criteria used to define melatonin exposure (≥90 DDD within 1 year) may still allow for inclusion of intermittent or irregular users. The authors should justify this threshold and discuss how variations in adherence might affect the results. Furthermore, providing subgroup analyses by dose intensity or duration would clarify whether a dose-response relationship exists.
ANS: Thank you for this valuable comment.
We have clarified and discuss how variations in adherence might affect the results. in the Methods section, we add
“To further ensure adequate exposure, we excluded individuals who did not receive a second prescription for melatonin or a hypnotic BZD at least three months after the in-itial prescription[25]. This criterion, informed by prior pharmacoepidemiologic studies, was designed to minimize the inclusion of sporadic users and better capture those with sustained or repeated use[26]. While it does not confirm daily adherence, it offers a reasonable proxy for long-term exposure during the observation period and helps ex-clude patients with only short-term use.” (Line 113~119)
Due to the limitations of the TriNetX database, we were unable to conduct dose-response or duration-stratified analyses. We have mentioned this as in limitation section.
“Second, due to limitations of the database, we were unable to assess the dosage, formulation and duration of melatonin use, nor could we evaluate medication adherence. These limitations may have resulted in exposure misclassification and could potentially bias the estimated effects. Future studies should capture detailed treatment information to confirm these findings.”(Line 295~298)
Reviewer Comment: Potential Residual Confounding: Although IPTW was applied and baseline characteristics were balanced, residual confounding cannot be ruled out—especially given that important cataract risk factors such as sunlight exposure, BMI, or smoking were not available in the database. The authors acknowledge this but could elaborate on how such unmeasured confounders might bias the effect estimates.
ANS: We thank the reviewer for this important comment. We agree that residual confounding remains a concern in observational studies, especially when lifestyle or environmental risk factors (e.g., UV exposure, smoking, BMI). Our propensity score matching model included adjustments for body mass index (BMI) and nicotine dependence as proxies for certain lifestyle-related factors. However, we acknowledge that other unmeasured variables, particularly ultraviolet (UV) exposure, were not available in the database. We have now addressed this limitation in the revised manuscript. “First, as an observational study, it may still be subject to coding errors or confounding biases. Although we selected an active comparator group and employed PSM to minimize these influences, the possibility of residual bias cannot be entirely ruled out, especially from unmeasured factors like ultraviolet (UV) exposure.”(Line 291~294)
Reviewer Comment: Cataract Surgery as a Surrogate for Cataract Incidence: Using cataract surgery as a surrogate outcome for cataract incidence is acceptable but should be interpreted with caution. Surgical decisions may depend on accessibility, patient preferences, or physician judgment. The authors should acknowledge the limitation that not all cataracts result in surgery and that melatonin might influence the timing or threshold for surgery, rather than the biological development of cataract itself.
ANS: We sincerely appreciate the reviewer’s insightful comment. We fully agree that the use of cataract surgery as a surrogate for cataract incidence should be interpreted with caution, given the influence of non-clinical factors such as patient preference, physician judgment, and healthcare access. We would like to clarify that in our study, cataract outcomes were identified based on ICD-10 diagnostic codes, which reflect clinically diagnosed cataracts rather than surgical events. This approach allows for the inclusion of a broader spectrum of cataract cases, including those not yet progressing to surgery. We add “Fifth, Cataract outcomes were based on ICD-10 diagnosis codes, which help capture non-surgical cases but may not fully reflect disease severity or progression.” (Line 306~308) in limitation section.
Treatment Duration and Dose Comparability: The manuscript lacks a direct comparison of treatment duration or cumulative exposure between the melatonin and BZD groups. While the exposure definition (≥90 DDD within 1 year) establishes a threshold, it does not reflect actual usage patterns. Given that melatonin may be used intermittently or seasonally, and BZDs may be prescribed for longer periods (often chronically), this difference in exposure duration may confound the observed outcomes. The authors should provide descriptive statistics on the actual treatment duration and cumulative dose (if available) to clarify whether the groups are comparable in terms of exposure intensity and chronicity.
ANS: We appreciate the reviewer’s thoughtful observation regarding potential differences in treatment duration and cumulative exposure between the melatonin and BZD groups. Due to the inherent limitations of the TriNetX platform, we were unable to access detailed individual-level data on treatment duration. We fully acknowledge that melatonin may be used intermittently or seasonally, while BZDs are often prescribed continuously. This difference may contribute to exposure misclassification and could potentially influence outcome estimates. We have now discussed this limitation more explicitly in the revised Discussion section (paragraph 6):
“Second, due to limitations of the database, we were unable to assess the dosage, formulation and duration of melatonin use, nor could we evaluate medication adherence. These limitations may have resulted in exposure misclassification and could potentially bias the estimated effects.”(line 295~299)
Minor Comments
Abstract: Consider clarifying that the comparator group consisted of patients with sleep disorders treated with BZDs to avoid confusion.
ANS: Revised as suggested. The abstract now reads:
“Adults aged 40 years or older with a diagnosis of sleep disorder who initiated treatment with either melatonin or hypnotic benzodiazepines (BZD) were included. Patients receiving hypnotic BZD therapy constituted the active comparator group.”(Line 30~33)
Introduction: Expand slightly on the role of oxidative stress in cataractogenesis and how melatonin’s pharmacodynamics might counteract these effects.
Ans:
We add “Oxidative stress drives cataractogenesis by promoting protein aggregation, lens opacification, and epithelial cell apoptosis. Melatonin’s potent antioxidant and anti-inflammatory effects counteract these processes by scavenging reactive oxygen species and stabilizing mitochondrial function.”(Line 56~60) in introduction.
Figures and Tables: Correct Figure 3: Outomce should be outcome
ANS: Corrected. Thank you for pointing this out.
Once again, we are truly thankful for your valuable comments, which have greatly improved the rigor and clarity of our manuscript. We hope that the revisions meet your expectations, and we look forward to your favorable consideration.
Round 2
Reviewer 1 Report
The study is novel. Though based on prior studies on diabetic patients and Age-related macular degeneration subjects the findings are along expected lines. Given the novelty of studies, the article is well written. However, at authors discretion a light professional English editing will be useful. Author are recommended to cite the following relevant meeting abstract available online: https://iovs.arvojournals.org/article.aspx?articleid=2787085 Another paper relevant for addition in discussion: AAO URL=https://www.aao.org/education/editors-choice/melatonin-use-may-be-linked-with-lower-risk-of-
The study is novel. Though based on prior studies on diabetic patients and Age-related macular degeneration subjects the findings are along expected lines. Given the novelty of studies, the article is well written. Though the authors have performed light editing however, at authors discretion a light professional English editing will be useful. Some phrases used can be improved. At some places professional English editing will allow brevity and increase clarity. This is too frequent to point out page and line numbers. However, their work is understandable even without edits. Editing will improve the presentation and clarity but current manuscript is fine even without the same.
Author are recommended to cite the following relevant meeting abstract available online: https://iovs.arvojournals.org/article.aspx?articleid=2787085; Another paper relevant for addition in discussion: AAO URL=https://www.aao.org/education/editors-choice/melatonin-use-may-be-linked-with-lower-risk-of-
Author Response
Dear Editor and Reviewer,
We sincerely thank the reviewer for the positive feedback on the novelty and overall quality of our manuscript. We appreciate your suggestion regarding professional English editing to improve the clarity and presentation. In response, we have carefully revised the manuscript for language and style, and have additionally sought assistance with professional English editing to enhance fluency, brevity, and readability throughout the text.
Reviewer comment: The study is novel. Though based on prior studies on diabetic patients and Age-related macular degeneration subjects the findings are along expected lines. Given the novelty of studies, the article is well written. Though the authors have performed light editing however, at authors discretion a light professional English editing will be useful. Some phrases used can be improved. At some places professional English editing will allow brevity and increase clarity. This is too frequent to point out page and line numbers. However, their work is understandable even without edits. Editing will improve the presentation and clarity but current manuscript is fine even without the same.
Reply: We sincerely thank the reviewer for the positive feedback on the novelty and overall quality of our manuscript. We appreciate your suggestion regarding professional English editing to improve the clarity and presentation. In response, we have carefully revised the manuscript for language and style and have additionally sought assistance with professional English editing to enhance fluency, brevity, and readability throughout the text.
Reviewer comment: Author are recommended to cite the following relevant meeting abstract available online: https://iovs.arvojournals.org/article.aspx?articleid=2787085; Another paper relevant for addition in discussion: AAO URL=https://www.aao.org/education/editors-choice/melatonin-use-may-be-linked-with-lower-risk-of-
Reply: We appreciate your valuable recommendations for citing additional relevant literature. We have now cited the ARVO meeting abstract to strengthen the contextual background of our study (line 76-78). As for the AAO Editors’ Choice article, we would like to note that we have already cited and discussed the full peer-reviewed version of that study in our manuscript (line 258-261).
We believe these changes have improved the overall quality and completeness of the manuscript. All modifications have been highlighted in the revised version.

Reviewer 2 Report
The authors have adequately addressed all my comments. I therefore recommend acceptance of the manuscript in its current form.
All my previous comments have been addressed, and I have no further suggestions.
Author Response
Reviewer comment: The authors have adequately addressed all my comments. I therefore recommend acceptance of the manuscript in its current form.
Reply: We sincerely thank the reviewer for their positive feedback and recommendation for acceptance. We greatly appreciate your valuable time and constructive comments, which have helped us improve the manuscript.
Reviewer 3 Report
This study makes a timely and valuable contribution to the growing literature on oxidative stress and ocular disease, particularly cataracts. While melatonin’s antioxidant and anti-inflammatory properties have been well documented in preclinical models, clinical evidence of its protective role in humans has been limited. By analyzing real-world data through a robust target trial emulation design, this study offers the first large-scale epidemiological evidence suggesting that melatonin use may reduce the risk of cataract development.
The authors effectively bridge experimental data with population-level outcomes, supporting the plausibility of melatonin as a preventive agent in cataractogenesis. Their careful methodology, including active comparator design, propensity score matching, and sensitivity analyses, enhances the reliability of the findings despite the observational nature of the data.
Overall, this work lays important groundwork for future clinical trials and advances the case for melatonin’s potential role in ocular health maintenance.
The authors have submitted a thoroughly revised version of their manuscript, which investigates the association between melatonin use and the risk of cataract development through a target trial emulation design using the TriNetX electronic health records database. I appreciate the detailed and thoughtful responses to the previous round of reviewer comments. After a careful reading of the revised manuscript, the supplementary materials, and the cover letter detailing the point-by-point replies, I find that the authors have adequately and convincingly addressed all the key concerns raised during the initial review.
One of the central critiques in the previous round pertained to the comparator group choice—specifically, the use of benzodiazepine (BZD) users as the control. While this was originally justified on the grounds of indication similarity (i.e., treatment of sleep disorders), it was correctly pointed out that BZDs may have pro-oxidant properties, potentially exaggerating the apparent protective effect of melatonin. To address this, the authors have now included a sensitivity analysis comparing melatonin users to a non-BZD, non-melatonin user group. This new analysis demonstrates a statistically significant reduction in cataract risk among melatonin users (HR 0.785 for age-related cataract and 0.885 for other cataracts), strengthening the interpretation that melatonin may have an independent protective effect, rather than merely appearing safer than BZDs. These results are clearly presented in Table S4 and referenced appropriately in both the methods and results sections.
Another major concern was the lack of biochemical or biological endpoints, such as oxidative stress markers, to substantiate the hypothesized antioxidant mechanism of melatonin. Given the inherent limitations of claims and EHR databases like TriNetX, direct measurement of such biomarkers was understandably not feasible. However, the revised manuscript now provides a stronger biological rationale by incorporating recent and relevant preclinical literature. These studies support melatonin’s ability to reduce malondialdehyde (MDA) levels and enhance antioxidant enzyme activity (e.g., SOD, GSH-Px), as well as modulate Nrf2 and inflammasome pathways in lens tissue. This addition improves the contextual grounding of the observed epidemiological associations.
Regarding the exposure definition, the prior concern was that the threshold of ≥90 defined daily doses (DDDs) over one year might include irregular or short-term users. The authors have now added an additional criterion: patients must have received a second prescription at least three months after the first, aligning with accepted pharmacoepidemiologic methods to identify sustained use. Furthermore, they acknowledge limitations in assessing adherence, dose intensity, and seasonal variation in melatonin use. This is transparently discussed in the limitations section, and while these constraints cannot be fully overcome within this dataset, the authors have mitigated their potential impact as best as possible.
The issue of residual confounding—particularly by factors not captured in the database, such as sunlight exposure—was also thoughtfully revisited. The authors note that their propensity score model did adjust for available proxies (e.g., nicotine dependence, BMI), and explicitly acknowledge the potential influence of unmeasured variables such as UV exposure. This open discussion of limitations adds to the credibility of the analysis.
The reviewer also previously raised concerns regarding the validity of using cataract surgery as a surrogate endpoint. In response, the authors clarified that cataract incidence was identified via ICD-10 diagnostic codes, not surgical records, thereby including non-surgical cataracts and mitigating concerns related to variations in surgical decision-making. This is a key clarification that enhances the interpretability of the primary outcome.
Finally, the revised manuscript corrects minor issues noted in the prior version—clarifying the abstract to define the comparator group more precisely, fixing a typographical error in a figure title, and expanding the introduction to better explain melatonin’s biological role in ocular tissues and its relevance to oxidative stress.
Overall, the revised manuscript is substantially improved. The authors have demonstrated diligence and scientific integrity in their responses and revisions. The additional analyses, expanded discussion, and clear acknowledgment of study limitations all contribute to a more balanced and convincing presentation of their findings. While residual limitations inherent to observational database studies remain, they have been appropriately recognized and discussed.
Recommendation: Accept for publication.
Author Response
Reviewer comment: This study makes a timely and valuable contribution to the growing literature on oxidative stress and ocular disease, particularly cataracts. While melatonin’s antioxidant and anti-inflammatory properties have been well documented in preclinical models, clinical evidence of its protective role in humans has been limited. By analyzing real-world data through a robust target trial emulation design, this study offers the first large-scale epidemiological evidence suggesting that melatonin use may reduce the risk of cataract development.
The authors effectively bridge experimental data with population-level outcomes, supporting the plausibility of melatonin as a preventive agent in cataractogenesis. Their careful methodology, including active comparator design, propensity score matching, and sensitivity analyses, enhances the reliability of the findings despite the observational nature of the data.
Overall, this work lays important groundwork for future clinical trials and advances the case for melatonin’s potential role in ocular health maintenance.
Reply: We sincerely thank the reviewer for the positive and encouraging feedback. We are pleased that our study’s novelty, methodological rigor, and contribution to bridging preclinical and real-world evidence have been recognized.